# Selection and Validation of Reference Genes for RT-qPCR Analysis of Gene Expression in *Nicotiana benthamiana* upon Single Infections by 11 Positive-Sense Single-Stranded RNA Viruses from Four Genera

**DOI:** 10.3390/plants12040857

**Published:** 2023-02-14

**Authors:** Ge Zhang, Zhuo Zhang, Qionglian Wan, Huijie Zhou, Mengting Jiao, Hongying Zheng, Yuwen Lu, Shaofei Rao, Guanwei Wu, Jianping Chen, Fei Yan, Jiejun Peng, Jian Wu

**Affiliations:** 1State Key Laboratory for Managing Biotic and Chemical Threats to the Quality and Safety of Agroproducts, Institute of Plant Virology, Ningbo University, Ningbo 315211, China; 2Key Laboratory of Biotechnology in Plant Protection of MARA and Zhejiang Province, Institute of Plant Virology, Ningbo University, Ningbo 315211, China; 3Hunan Plant Protection Institute, Hunan Academy of Agricultural Sciences, Changsha 410125, China

**Keywords:** RT-qPCR, reference gene, positive-sense single-stranded RNA, *Nicotiana benthamiana*

## Abstract

Quantitative real-time PCR (RT-qPCR) is a widely used method for studying alterations in gene expression upon infections caused by diverse pathogens such as viruses. Positive-sense single-stranded (ss(+)) RNA viruses form a major part of all known plant viruses, and some of them are damaging pathogens of agriculturally important crops. Analysis of gene expression following infection by ss(+) RNA viruses is crucial for the identification of potential anti-viral factors. However, viral infections are known to globally affect gene expression and therefore selection and validation of reference genes for RT-qPCR is particularly important. In this study, the expression of commonly used reference genes for RT-qPCR was studied in *Nicotiana benthamiana* following single infection by 11 ss(+) RNA viruses, including five tobamoviruses, four potyviruses, one potexvirus and one polerovirus. Stability of gene expression was analyzed in parallel by four commonly used algorithms: geNorm, NormFinder, BestKeeper, and Delta CT, and RefFinder was finally used to summarize all the data. The most stably expressed reference genes differed significantly among the viruses, even when those viruses were from the same genus. Our study highlights the importance of the selection and validation of reference genes upon different viral infections.

## 1. Introduction

Infections of positive-sense single-stranded RNA (ss(+) RNA) viruses are major threats to agricultural production [1]. To systemically infect plant cells, ss(+) RNA viruses cooperate with host factors to manipulate host genes, thereby establishing a suitable microenvironment for viral replication and trafficking. Therefore, analysis of gene expression upon viral infection is critical for the identification of potential anti-viral factors. To study virus–host interactions, fluorescence-based quantitative real-time PCR (RT-qPCR) is often used to quantify the transcript levels of target genes [2,3]. The use of qPCR to accurately quantify transcript abundance is largely influenced by the amount of initiating material, RNA quality, reverse transcriptase efficiency, amplification efficiency, etc. [4]. To normalize these differences, the use of stably expressed reference genes is required [5]. The internal reference gene is an internal reaction control whose sequence is different from the target gene. To serve as a reliable reference for RT-qPCR a gene must meet several important criteria, of which the most important is that its expression level is not affected by any of the experimental treatments [6]. In addition, its expression should vary minimally in different tissues as well as under different physiological conditions. *Actin*, *GAPDH* and *EF1α* play critical roles in life activities, and their expression is largely independent of external factors such as abiotic stresses [7]. Therefore, these genes are widely used as references in RT-qPCR to analyze gene expression [8]. In *Nicotiana benthamiana* (*N. benthamiana*), the stability of several reference genes has been tested in developmentally distinct tissues and during abiotic stresses [9]. Moreover, previous studies have also analyzed the stability of certain reference genes during infection with a variety of viruses, including necroviruses, benyviruses, hordeiviruses, potexviruses, and tobacco rattle virus [10,11]. However, the stability of expression of most reference genes in host plants during infection with most of the reported plant viruses has not been well studied.

It is known that the expression of some genes that can be used as references under certain conditions may show significantly altered expression under other conditions [7]. To systemically infect hosts, viruses replicate and assemble in host cells using host components to produce new progeny viruses and initiate a new round of life cycle [12]. In this process, plant viruses typically interfere with and engage the host’s cellular pathways, and a large portion of the most widely used reference genes are components of these cellular pathways [13]. In addition, different viral infestations may affect different pathways, and the extent to which these pathways are affected varies depending on the time of viral infestation and the type of cells infected. Some ss(+) RNA viruses may hijack host metabolic enzymes and housekeeping proteins to facilitate their replication. For instance, tomato bushy stunt virus (TBSV) can bind *GAPDH*, a commonly used reference gene that plays critical roles in glycolysis [14], membrane fusion, vascular bundles, nuclear RNA output, DNA replication, DNA repair [15] and RNA stability [16]. By binding GAPDH to the 3′ untranslated region (UTR) of the TBSV negative-sense strand during replication the synthesis of the positive-sense chain is stimulated by increasing the stability of template [17,18]. However, the stability of GAPDH expression during TBSV infection has not been carefully analyzed in many host plants. Another example is that tobacco mosaic virus (TMV) can bind *eEF1A*, a highly conserved and abundant housekeeping protein in eukaryotes [19], to maintain the activity of viral RdRp [20]. Moreover, TMV is known to induce the expression of *EF1A* [21]. Therefore, it is particularly important to select and validate RT-qPCR reference genes to analyze the expression of host genes upon viral infection [22].

Owing to its ability to be infected by a wide variety of plant viruses, *N. benthamiana* has been widely used as an experimental model for studying plant–virus interactions [23]. Plant virus vector-based expression systems, the construction of virus-induced gene silencing (VIGS) systems, and agrobacterium infiltration are all developed using *N. benthamiana* plants [24]. In this study, *N. benthamiana* infection was therefore used as a model to analyze the stability of commonly used reference genes in the context of infections caused by 11 ss(+) RNA viruses, including five tobamoviruses (TMV, tomato brown rugose fruit virus (ToBRFV), pepper mild mottle virus (PMMoV), tobacco mild green mosaic virus (TMGMV) and tomato mottle mosaic virus (ToMMV)), four potyviruses (turnip mosaic virus (TuMV), chilli ringspot virus (ChiRSV), chilli veinal mottle virus (ChiVMV) and pepper mottle virus (PePMoV)), one potexvirus (potato virus X (PVX)) and one polerovirus (pod pepper vein yellows virus (PoPeVYV)). Five commonly used algorithms, geNorm, NormFinder, BestKeeper, Delta CT, and RefFinder were adopted to identify the most stably expressed reference genes following infection by the different viruses. The results show that the most promising reference genes can even be different for viruses of the same genus and identify useful reference genes for future studies of gene expression following infection by these 11 viruses. Our analysis also revealed that the selection and validation of reference genes in the context of different viral infections are necessary.

## 2. Results

### 2.1. Assessment of Primer Specificity and Amplification Efficiency

A total of 28 reference genes and their forward and reverse primers used for RT-qPCR were obtained from published literature. The primer sequences of all genes and information on the length of amplification products are provided in Appendix A. We used RT-qPCR melting curve analysis, 1% agarose gel electrophoresis and ethidium bromide staining, and PCR product sequencing to verify the specificity of the primers. Among all the 28 selected candidate genes in Appendix A, genes in black letters indicate that the melting curve obtained by RT-qPCR was single-peaked and only one band was visualized after ethidium bromide staining (Figure 1A and Appendix A), indicating high specificity of the amplification product; genes in red letters indicate the presence of double peaks in the melting curve or a red curve (Figure 1B), revealing the presence of non-specific amplification for that primer or low transcript levels of the target genes; genes in blue letters indicate that, as revealed by Sanger sequencing of the RT-qPCR products after cloning into pGEM^®^-T vector, the obtained sequence of at least three clones did not match the expected target gene sequences. We finally selected 13 genes from the 28 candidates for the subsequent experiments. The amplification efficiencies of these 13 primer pairs were analyzed. The results showed that the amplification efficiencies ranged from 91.9% to 110% (Table 1), which were all within the acceptable range of 90–110%. Furthermore, the standard curves demonstrated good linear relationships (R^2^ > 0.9874) between the cycle threshold (Ct) values and the log-transformed copy numbers of 13 tested reference genes (Table 1). The above results indicate that the primers of these 13 pairs of candidate reference genes were suitable for following RT-qPCR experiments.

### 2.2. Confirmation of Viral Infection and Sampling

At 6 days after inoculation, plants infected with six viruses (TMV, ToBRFV, PMMoV, TMGMV, ToMMV, and TuMV) had obvious symptoms (curved leaves, yellow spotting, etc.) in the systemic leaves, whereas it took 15 days for plants infected with the other five viruses to show typical symptoms. Symptoms induced by infections of representative viruses from four genera are shown in Appendix A. The data of RT-PCR confirmation of viral infection are shown in Appendix A.

### 2.3. Expression Analysis of the 13 Selected Reference Genes by RT-qPCR

To analyze the differential expression of the 13 candidate reference genes in *N. benthamiana* under infection of each of the 11 viruses, RNA samples were prepared using leaf samples collected at either 6 or 15 dpi depending on the time symptoms appeared (see above). RNA integrity was analyzed by 5% urea polyacrylamide gel electrophoresis. After ethidium bromide staining, ribosomal RNA bands with sharp edges were observed, indicating high RNA integrity (Appendix A). RNA concentrations were determined and only the ones with a 260/280 ratio close to 2.0 and a 260/230 ratio within the range of 2.0–2.2 were used in RT-qPCR. Prior to reverse transcription, genomic DNA was digested by incubating the RNA samples with DNase I for 2 h at 37 °C. Melting curves of the RT-qPCR products were checked to make sure only a single sharp peak was obtained. Ct values of three biological replicates of each virus-infected group and the Mock infection group were displayed by box line plots to represent the variations (Figure 2).

Among the tobamoviruses, the greatest variation in gene expression was in *GAPDH* after infection by TMV, TMGMV, ToBRFV and ToMMV. Following PMMoV infection, Ct values of gene *AGO2* varied the most. Gene *Lip* showed the smallest variations in Ct values after infection by TMV or ToBRFV; Ct values of gene *Tspan* varied the least in the case of TMGMV and ToMMV single infection; for PMMoV infection, the gene with the least variation in Ct value was *KLC.*

Among the four potyviruses TuMV, ChiRSV, ChiVMV, and PePMoV, the genes that varied the most in Ct values were *GAPDH*, *AGO2*, *AGO2*, and *GADPH*, respectively; the genes with the least variations in Ct values were *RdR6*, *UBC*, *Lip*, and *Lip*, respectively.

In the case of PVX infection (genus *Potexvirus*), the Ct value of gene *Lip* varied the most and the gene *Tspan* varies the least and for PoPeVYV (genus *Polerovirus*) the gene with the biggest variation in Ct value was *L23* and the gene with the smallest variation was *UBC*.

### 2.4. Stability Analysis of the 13 Selected Reference Genes

To further evaluate the expression stability of the 13 candidate reference genes, four different algorithms (geNorm, NormFinder, BestKeeper, and Delta-CT) were used to calculate expression stabilities individually. Finally, a comprehensive analysis platform RefFinder was used to integrate the values obtained from the four algorithms to identify the one with the highest stability upon the infection of each virus.

As shown in Table 2, the different algorithms were not consistent with one another in identifying the most and least stable genes for each virus. When integrated in RefFinder (Table 3), the top four reference genes for TMV were *Lip, eIF4A, EF1α,* and *L23* while *GAPDH*, *AGO2*, and *ACT* were the least stable; since the M value of GAPDH was greater than 1.5 it is very unsuitable as a reference in this regard. For PMMoV infection, RefFinder identified the top four stable reference genes as *RdR6*, *UBC*, *Tspan*, and *ACT* while the least stable were *AGO2*, *PGK*, and *L23*, none of which were suitable as a reference gene. For TMGMV, *F-BOX*, *RdR6*, *Lip*, and *ACT* were the top four stable reference genes while *GAPDH*, *AGO2*, *UBC*, and *eIF4A* were not suitable as reference genes. For ToBRFV, the top four stable reference genes were *F-BOX*, *UBC*, *EF1α*, and *PGK,* while *AGO2*, *GAPDH,* and *L23* were the three least stable genes. For ToMMV, *RdR6*, *Tspan*, *Lip*, and *eIF4A* were the top four stable genes while the least stable were *GAPDH*, *ACT*, *AGO2*, and *L23*.

Among the potyviruses, RefFinder identified *PGK, RdR6, eIF4A*, and *Tspan* as the top four stable reference genes for TuMV, while the most unstable were *GAPDH*, *AGO2*, *ACT*, and *UBC* (Table 3). For ChiRSV, *Tspan*, *UBC*, *EF1α*, and *RdR6* were the top four stable reference genes, while the most unstable were *AGO2*, *L23*, *ACT*, and *GAPDH*. For ChiVMV, *Tspan*, *eIF4A*, *UBC*, and *F-BOX* were the four most stable genes while the most unstable were *AGO2*, *GAPDH, L23*, and *RdR6*). For PePMoV, the four algorithms gave very similar results with *Lip*, *eIF4A, KLC*, and *Tspan* being the most stable (Table 2); the most unstable were *AGO2*, *GAPDH*, *ACT*, and *RdR6* (Table 3).

For PVX, RefFinder identified *F-BOX*, *Tspan*, *ACT*, and *PGK* as the top four stable candidate reference genes, while the most unstable were *Lip, KLC, EF1α*, and *RdR6*. For PoPeVYV, the four most stable genes were *Tspan*, *F-BOX*, *eIF4A*, and *PGK*, with *RdR6*, *AGO2*, *L23*, and *Lip* being the most unstable.

## 3. Discussion

RT-qPCR has become one of the important methods for rapid, sensitive, and quantitative comparison of gene expression levels [25]. Many experimental factors, such as the quantity of starting material, the quality of RNA, and the efficiency of reverse transcriptase, can affect the efficacy of RT-qPCR [4,26]. Therefore, accurate normalization, which relies on the selection of housekeeping reference genes, is crucial. Unfortunately, there are no universal references that can be used under diverse conditions to accurately reflect alterations in gene expression [27]. In the case of different viral infections that can globally affect gene expression, it is particularly important to select and validate suitable internal references to identify the targets of viruses and potential anti-viral factors [28]. Therefore, a systematic evaluation of reference genes must be performed before the analysis of gene expression upon viral infections [29].

Interestingly, our comprehensive data show that different reference genes should be used for different viruses, even when those viruses are members of the same genus. For example, while *Lip*, *eIF4A, EF1α,* and *L23* appeared to be most suitable as reference genes for infection by TMV, those most suitable for PMMoV (another tobamovirus) were *RdR6*, *UBC*, *Tspan,* and *ACT*. Thus, there is no single gene appropriate for use with all tobamoviruses. Similar conclusions were obtained after the analysis of gene expression in the context of infections by four different potyviruses. Our study therefore suggests that reference genes need to be evaluated and selected on a case-by-case basis in the context of viral infections; they cannot be simply adopted from previous studies using a healthy plant or a different virus.

Our conclusions are also supported by previous studies. *F-BOX* is stably expressed in a variety of plant species under different biotic or abiotic stresses. In *Arabidopsis thaliana, F-BOX* is one of the genes that exhibits the most stable expression levels when plants are infected by viruses [30] or when they are under metal stress [31]. Analysis of gene expression in *N. benthamiana* plants infected with five RNA plant viruses also revealed that the *F-BOX* was the most stable reference gene [10]. In our study, *F-BOX* was also the most suitable reference gene in *N. benthamiana* plants infected with TMGMV, ToBRFV, and PVX. However, F-BOX expression levels varied a lot in plants infected by TMV or TuMV. Tetraspanins are a family of small transmembrane proteins that play crucial roles in intracellular transport, cellular signal transduction, and cell migration [32]. More than 33 Tspans, such as *Tspan7*, *span9*, *CD9*, *CD81*, and *CD151*, are known to be involved in the process of virus infestation in human cells [33]. Tetraspanins in *A. thaliana* facilitate CMV infection through specific membrane-recognition motifs [34]. However, in our study *Tspan* (Tetraspanin-20) displayed stable expression in *N. benthamiana* plants infected by ToMMV, ChiRSV, ChiVMV or PVX, suggesting that *Tspan* plays no significant role in infection by these viruses. GAPDH, AGO2, and actins are three of the most widely used reference genes in various plant species [22,35], but they have altered expression under various biotic or abiotic stresses [26,30,36]. Consistently, our study also revealed the altered expression of *GAPDH*, *AGO2*, and actins upon viral infections, suggesting that these are not suitable as reference genes in studies of viral infection. Interestingly, a previous study reported that *L25* and *EF-1α* can be used as stable reference genes for gene expression analysis in both developmental and stress-treated samples of *N. benthamiana* plants [9]. The complexity of gene expression analysis under biotic stress treatments, especially viral infections, was highlighted by that study and our results.

## 4. Materials and Methods

### 4.1. Plant Materials and Virus Inoculation

*N. benthamiana* plants were grown in a growth chamber with a 16-h light/8-h dark cycle at 24 °C and 50% relative humidity. Infectious clones of 11 viruses, including TMV, PMMoV, TMGMV, ToBRFV, ToMMV, TuMV, ChiRSV, ChiVMV, PePMoV, PVX, and PoPeVYV were all from our laboratory. The information of genome size, backbone vector, and promoter were listed in Appendix A. All viruses were agro-inoculated into *N. benthamiana* at the five-true-leaf stage according to published methods [37]. Three biological replicates (three plants per replicate) were included for each virus and mock control. Viral infection was confirmed by disease symptoms.

### 4.2. Total RNA Isolation and cDNA Synthesis

Depending on the onset of symptoms (see Results Section 2.2, above), leaf samples were collected from the first two fully expanded systemic leaves of *N. benthamiana* plants infected with TMV, ToBRFV, PMMoV, TMGMV, ToMMV, and TuMV at 6 days post inoculation (dpi) and from those infected with the remaining five viruses and 15 dpi. Total RNA was extracted using TRIzol^TM^ Reagent (Invitrogen, Shanghai, China) according to the manufacturer’s instructions. RNA concentration was measured using a Nanodrop ONE spectrophotometer (Nanodrop Technologies, Rockland, DE, USA). RNA integrity was analyzed by 5% urea polyacrylamide gel electrophoresis, according to standard procedures [38]. First Strand cDNA was synthesized by reverse transcription (RT) using EasyScript^®^. All-in-One First-Strand cDNA Synthesis SuperMix for qPCR (Transgen, Beijing, China) with 1 μg total RNA as template. All procedures were completed following the manufacturer’s protocol. Thermal conditions for RT were: 15 min at 37 °C, and then 5 s at 85 °C to inactivate the enzyme.

### 4.3. Selection of Candidate Reference Genes

Reference genes and the forward and reverse primers were obtained from published literature [10,39,40,41,42,43,44], or designed in this study. Primer sequences and the references are listed in Appendix A. Primer specificity was evaluated using RT-qPCR and PCR amplification, in which the cDNA sample was derived from an RNA sample from a healthy *N. benthamiana* plant. The amplified PCR products were visualized by ethidium bromide staining after 1% agarose gel electrophoresis. PCR products were also purified and cloned into blunt zero T-vector (Transgen, Beijing, China) and sequenced. The amplification efficiency of each pair of primers was analyzed using the built-in software of the Roche LightCycler480 instrument and EXCEL. Standard curves were generated from a five-fold dilution (initial concentration = 14.6 ng/μL) series of one sample.

### 4.4. RT-PCR Confirmation of Viral Infection and RT-qPCR

RT-PCR was performed to confirm the existence of viral genome RNA in systemic leaves. Random primers were used to synthesize cDNA through RT. Primer sequences of PCR were listed in Appendix A. RT-qPCR was performed on a LightCycler480 instrument (Roche, Switzerland) with 384-well PCR plates. Reaction mixtures were composed of 18 μL of ChamQ Universal YBR QPCR Master Mix (Vazyme, Nanjing, China), 0.6 μL of each primer (10 μM), 6 μL of cDNA template, and 10.8 μL of RNase free water. Three biological replicates were included for each experiment, and three technical replicates were set for each biological replicate. Thermal cycles of RT-qPCR were: 95 °C for 30 s, then 40 cycles of 15 s at 95 °C, 15 s at 55 °C, and 30 s at 72 °C. Melting curves were prepared through the following thermal conditions: 95 °C for 10 s, 60 °C for 30 s, and 95 °C for 15 s. Amplification efficiencies were calculated using the built-in software of the Roche LightCycler 480 instrument.

### 4.5. Evaluation of the Expression Stability of Reference Genes Using GeNorm, NormFinder, Delta-CT, BestKeeper, and RefFinder

To analyze gene expression stability, four algorithms (geNorm [45], NormFinder [46], BestKeeper [47], and Delta-CT [48]) and an online comprehensive analysis platform RefFinder [49] were used following manufacturer’s instructions.

For geNorm and NormFinder, the raw Ct values were converted to the data input format (2^−ΔCt^) required by the software. Moreover, ΔCt = each Ct value—the lowest Ct value. In contrast, raw Ct values were directly used as input data in BestKeeper and RefFinder analysis. The output of geNorm software analysis is the expression stability value M, which represents the average pairwise variation between a single gene and other internal reference genes. With M = 1.5 as the cutoff value, a low M value indicates stable gene expression. In addition, geNorm also provides pairwise variation values (Vn/n+1) to represent the optimal number of reference genes for normalization. When Vn/n+1 < 0.15, the most suitable number of reference genes was n. When Vn/n+1 > 0.15, the most appropriate number of reference genes is n + 1. Normfinder provides the stability value SV to evaluate the differences in the expression of candidate reference genes and the variations among sample groups. Like the M value, a low SV value indicates stable gene expression. The Delta-CT method identifies the potential reference genes by comparing the relative expression between gene pairs. According to the lowest coefficient of variation (CV) and the lowest relative standard deviation (SD), BestKeeper software was used to determine the most stable reference genes. When SD > 1, the expression of the reference gene was unstable. RefFinder is a web-based comprehensive reference gene analysis platform. It integrates the values obtained using algorithms geNorm, NormFinder, BestKeeper, and Delta-CT to obtain the optimal internal reference genes.

## 5. Conclusions

In conclusion, our study identified optimal RT-qPCR reference genes for gene expression analysis in *N*. *benthamiana* plants upon single infections by 11 ss(+) RNA viruses from four genera. Our data also highlighted the importance of the selection and validation of reference genes for gene expression analysis in the context of viral infections.

## Figures and Tables

**Figure 1 plants-12-00857-f001:**
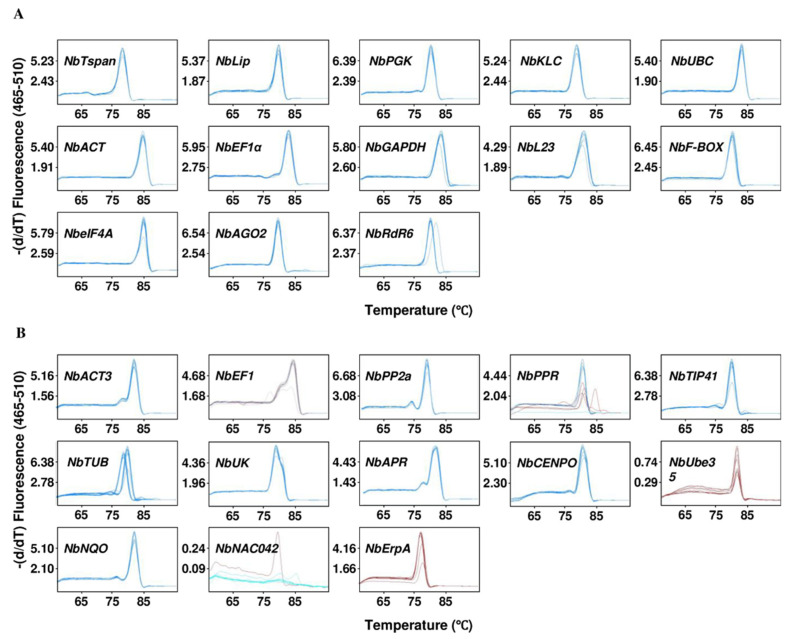
Melting curves of the RT-qPCR products of 28 reference genes tested in this study. (**A**) Melting curves of 13 genes showing a single peak. (**B**) Melting curves of the remaining 13 genes showing multiples peaks or red curves. Red curves indicate low RNA abundance in one or more replicates.

**Figure 2 plants-12-00857-f002:**
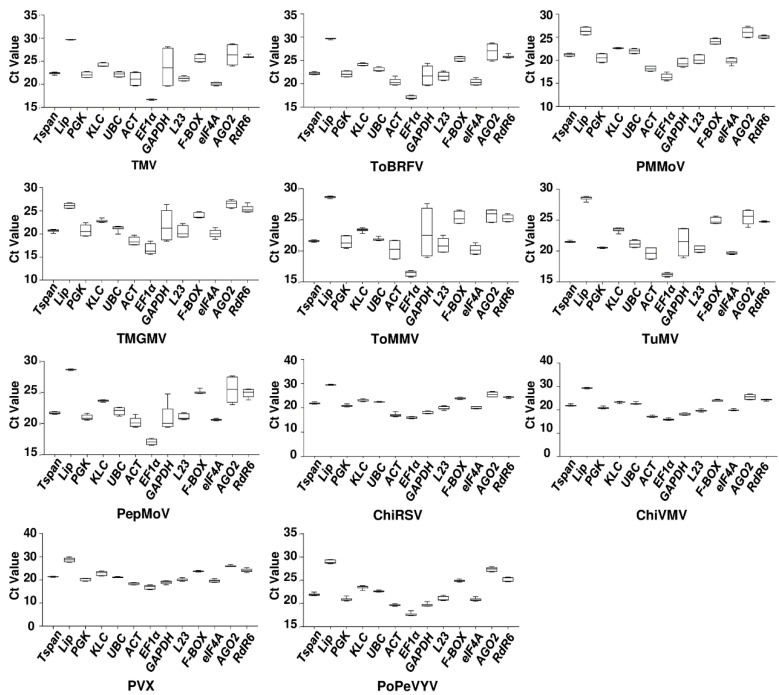
RT-qPCR Ct values of 13 candidate genes evaluated in the context of single infections of 11 viruses in *N. benthamiana*. Values are given as the mean Ct of three technical replicates of three biological replicates of each viral infection and Mock infection group. In total, six values are presented for each candidate reference gene. The same Mock group was used in all cases. Ct values are inversely proportional to the amount of template. The three lines of the box represent the 25th quartiles, median, and 75th quartiles. The whiskers represent minimum and maximum values.

**Table 1 plants-12-00857-t001:** Amplification efficiency of 13 pairs of primers.

Genes	Amplicon Length (bp)	E (%)	R^2^
*NbeIF4A*	135	107.4	0.996
*NbLip*	103	99.3	0.9963
*NbL23*	110	99.4	0.9951
*NbKLC*	81	106.4	0.996
*NbEF1α*	135	110	0.9874
*NbPGK*	105	105.5	0.9988
*NbTspan*	104	94.8	0.9961
*NbRdR6*	134	103.3	0.9960
*NbF-BOX*	125	96.1	0.9994
*NbUBC*	100	92.5	0.9956
*NbACT*	145	108	0.9914
*NbAGO2*	121	91.9	0.9923
*NbGAPDH*	125	96.1	0.9907

E: PCR amplification efficiency; R^2^: correlation coefficient.

**Table 2 plants-12-00857-t002:** Expression stability ranking of 13 candidate reference genes in the context of single infections by 11 viruses, as calculated by four algorithms.

Viruses	Gene	Algorithms								
		GeNorm		NormFinder		BestKeeper			Delta-CT	
		M	R	SV	R	CV	SD	R	SD	R
TMV	*NbeIF4A*	0.39	6	0.09	1	1.83	0.37	6	1.03	1
	*NbLip*	0.11	1	0.19	5	0.17	0.05	1	1.06	3
	*NbL23*	0.43	7	0.09	1	2.30	0.49	7	1.06	2
	*NbKLC*	0.34	5	0.16	4	1.40	0.34	5	1.06	4
	*NbEF1α*	0.11	1	0.28	7	0.46	0.08	2	1.08	5
	*NbPGK*	0.46	8	0.09	3	2.79	0.62	9	1.10	6
	*NbTspan*	0.21	3	0.33	8	0.70	0.16	3	1.11	7
	*NbRdR6*	0.26	4	0.46	9	0.74	0.19	4	1.15	8
	*NbF-BOX*	0.52	9	0.27	6	3.14	0.81	10	1.23	9
	*NbUBC*	0.59	10	1.00	10	2.46	0.55	8	1.42	10
	*NbACT*	0.73	11	1.12	11	6.57	1.39	11	1.69	11
	*NbAGO2*	1.07	12	3.08	12	8.50	2.24	12	3.11	12
	*NbGAPDH*	1.57	13	4.30	13	16.87	4.00	13	4.31	13
PMMoV	*NbeIF4A*	0.27	6	0.25	6	2.66	0.53	6	0.52	7
	*NbLip*	0.32	8	0.42	9	3.01	0.79	7	0.55	8
	*NbL23*	0.36	10	0.51	11	4.37	0.88	12	0.59	10
	*NbKLC*	0.41	12	0.40	8	0.48	0.11	1	0.65	12
	*NbEF1α*	0.29	7	0.28	7	4.12	0.67	11	0.49	5
	*NbPGK*	0.38	11	0.54	12	4.51	0.93	13	0.62	11
	*NbTspan*	0.10	1	0.09	4	1.43	0.30	3	0.50	6
	*NbRdR6*	0.10	1	0.05	1	1.36	0.34	2	0.49	4
	*NbF-BOX*	0.22	5	0.15	5	2.47	0.60	5	0.46	3
	*NbUBC*	0.18	3	0.05	1	2.35	0.52	4	0.42	1
	*NbACT*	0.20	4	0.05	1	3.12	0.57	8	0.43	2
	*NbAGO2*	0.62	13	1.73	13	3.72	0.97	9	1.74	13
	*NbGAPDH*	0.34	9	0.43	10	3.89	0.76	10	0.56	9
TMGMV	*NbeIF4A*	0.53	8	0.39	6	3.81	0.76	8	0.96	7
	*NbLip*	0.48	6	0.21	3	2.02	0.53	4	0.86	3
	*NbL23*	0.17	1	0.49	9	5.54	1.13	11	1.00	9
	*NbKLC*	0.57	9	0.47	8	1.08	0.25	1	0.91	6
	*NbEF1α*	0.23	3	0.25	5	5.75	0.96	12	0.91	5
	*NbPGK*	0.17	1	0.45	7	5.43	1.13	10	0.97	8
	*NbTspan*	0.67	10	1.01	10	1.09	0.22	2	1.18	10
	*NbRdR6*	0.41	5	0.15	2	2.51	0.64	7	0.83	1
	*NbF-BOX*	0.51	7	0.11	1	2.20	0.53	5	0.84	2
	*NbUBC*	0.74	11	1.09	11	2.02	0.43	3	1.26	11
	*NbACT*	0.37	4	0.23	4	4.50	0.83	9	0.89	4
	*NbAGO2*	0.84	12	1.47	12	2.30	0.61	6	1.54	12
	*NbGAPDH*	1.16	13	2.94	13	14.26	3.11	13	2.96	13
ToBRFV	*NbeIF4A*	0.19	3	0.33	7	3.14	0.64	10	0.80	6
	*NbLip*	0.48	9	0.29	6	0.33	0.10	1	0.82	7
	*NbL23*	0.40	7	0.66	11	4.10	0.89	11	0.97	9
	*NbKLC*	0.44	8	0.16	4	1.03	0.25	3	0.79	5
	*NbEF1α*	0.32	5	0.04	1	1.71	0.29	6	0.74	2
	*NbPGK*	0.13	1	0.28	5	2.80	0.62	8	0.78	4
	*NbTspan*	0.55	10	0.62	9	1.17	0.26	4	0.99	10
	*NbRdR6*	0.59	11	0.63	10	0.89	0.23	2	1.00	11
	*NbF-BOX*	0.13	1	0.06	3	2.08	0.53	7	0.74	1
	*NbUBC*	0.30	4	0.04	1	1.54	0.35	5	0.75	3
	*NbACT*	0.36	6	0.44	8	3.04	0.62	9	0.88	8
	*NbAGO2*	1.06	13	2.38	13	6.33	1.70	12	2.40	13
	*NbGAPDH*	0.82	12	2.12	12	9.52	2.07	13	2.14	12
ToMMV	*NbeIF4A*	0.46	7	0.13	2	3.53	0.71	7	0.90	2
	*NbLip*	0.20	1	0.82	10	0.45	0.13	2	1.09	9
	*NbL23*	0.62	10	0.40	6	5.36	1.12	11	1.10	10
	*NbKLC*	0.29	4	0.54	7	0.84	0.20	4	0.98	5
	*NbEF1α*	0.34	5	0.27	5	2.18	0.36	5	0.92	3
	*NbPGK*	0.53	8	0.15	4	4.07	0.87	8	0.96	4
	*NbTspan*	0.20	1	0.72	9	0.54	0.12	1	1.03	8
	*NbRdR6*	0.39	6	0.13	1	1.97	0.50	6	0.88	1
	*NbF-BOX*	0.57	9	0.14	3	3.61	0.92	10	0.99	6
	*NbUBC*	0.26	3	0.67	8	0.78	0.17	3	1.03	7
	*NbACT*	0.71	11	0.86	11	7.27	1.47	12	1.36	11
	*NbAGO2*	0.87	12	1.89	12	3.46	0.89	9	1.93	12
	*NbGAPDH*	1.29	13	3.62	13	16.14	3.70	13	3.63	13
TuMV	*NbeIF4A*	0.17	4	0.10	2	0.83	0.16	5	0.64	1
	*NbLip*	0.30	7	0.43	9	0.78	0.22	4	0.78	9
	*NbL23*	0.39	9	0.25	7	2.47	0.50	9	0.77	7
	*NbKLC*	0.26	6	0.14	5	1.10	0.26	6	0.70	5
	*NbEF1α*	0.22	5	0.11	4	1.70	0.27	7	0.71	6
	*NbPGK*	0.09	1	0.05	1	0.37	0.08	2	0.64	2
	*NbTspan*	0.12	3	0.18	6	0.45	0.10	3	0.67	4
	*NbRdR6*	0.09	1	0.10	3	0.29	0.07	1	0.66	3
	*NbF-BOX*	0.36	8	0.27	8	1.80	0.45	8	0.77	8
	*NbUBC*	0.47	10	0.88	11	2.67	0.56	10	1.04	10
	*NbACT*	0.55	11	0.76	10	4.45	0.87	11	1.05	11
	*NbAGO2*	0.70	12	1.61	12	4.46	1.13	12	1.64	12
	*NbGAPDH*	0.96	13	2.37	13	10.29	2.20	13	2.38	13
ChiRSV	*NbeIF4A*	0.15	1	0.35	9	1.96	0.39	8	0.52	7
	*NbLip*	0.33	8	0.34	7	0.48	0.14	2	0.56	8
	*NbL23*	0.47	12	0.62	12	2.87	0.57	11	0.70	12
	*NbKLC*	0.37	9	0.34	8	1.50	0.35	6	0.58	9
	*NbEF1α*	0.15	1	0.24	5	1.96	0.31	9	0.48	5
	*NbPGK*	0.23	4	0.25	6	1.51	0.32	7	0.47	3
	*NbTspan*	0.22	3	0.04	1	1.07	0.23	4	0.43	1
	*NbRdR6*	0.29	6	0.12	3	0.79	0.19	3	0.48	4
	*NbF-BOX*	0.31	7	0.17	4	1.23	0.29	5	0.49	6
	*NbUBC*	0.27	5	0.11	2	0.36	0.08	1	0.47	2
	*NbACT*	0.44	11	0.53	11	2.92	0.50	12	0.68	11
	*NbAGO2*	0.61	13	1.38	13	4.28	1.09	13	1.41	13
	*NbGAPDH*	0.40	10	0.52	10	2.59	0.47	10	0.66	10
ChiVMV	*NbeIF4A*	0.11	3	0.16	4	1.10	0.22	4	0.35	2
	*NbLip*	0.28	10	0.20	6	0.56	0.16	1	0.45	10
	*NbL23*	0.17	6	0.32	10	1.44	0.28	9	0.43	8
	*NbKLC*	0.25	9	0.20	5	0.94	0.22	3	0.44	9
	*NbEF1α*	0.19	7	0.15	3	1.87	0.30	11	0.39	5
	*NbPGK*	0.08	1	0.29	9	1.38	0.29	8	0.40	6
	*NbTspan*	0.12	4	0.14	2	1.13	0.25	5	0.34	1
	*NbRdR6*	0.33	12	0.32	11	0.88	0.21	2	0.51	11
	*NbF-BOX*	0.21	8	0.13	1	1.23	0.29	6	0.39	4
	*NbUBC*	0.08	1	0.24	7	1.30	0.29	7	0.38	3
	*NbACT*	0.15	5	0.28	8	1.63	0.28	10	0.40	7
	*NbAGO2*	0.49	13	1.33	13	4.32	1.10	13	1.34	13
	*NbGAPDH*	0.31	11	0.46	12	2.15	0.39	12	0.54	12
PePMoV	*NbeIF4A*	0.07	1	0.04	1	0.49	0.10	3	0.72	2
	*NbLip*	0.07	1	0.04	1	0.28	0.08	1	0.71	1
	*NbL23*	0.34	7	0.50	8	1.94	0.41	7	0.89	8
	*NbKLC*	0.12	3	0.07	3	0.48	0.11	2	0.73	3
	*NbEF1α*	0.41	8	0.45	7	2.71	0.46	10	0.89	7
	*NbPGK*	0.29	6	0.41	6	1.64	0.35	6	0.84	6
	*NbTspan*	0.14	4	0.08	4	0.68	0.15	4	0.74	4
	*NbRdR6*	0.53	10	0.63	10	2.50	0.62	8	1.00	10
	*NbF-BOX*	0.22	5	0.13	5	0.90	0.22	5	0.77	5
	*NbUBC*	0.48	9	0.59	9	2.52	0.55	9	0.96	9
	*NbACT*	0.59	11	0.83	11	3.16	0.64	11	1.10	11
	*NbAGO2*	1.07	13	2.36	13	8.10	2.06	13	2.39	13
	*NbGAPDH*	0.84	12	2.19	12	7.35	1.54	12	2.23	12
PVX	*NbeIF4A*	0.43	6	0.50	7	2.15	0.42	7	0.67	7
	*NbLip*	0.67	13	0.68	13	2.46	0.70	9	0.78	13
	*NbL23*	0.38	5	0.45	4	2.15	0.43	6	0.65	5
	*NbKLC*	0.65	12	0.67	12	3.59	0.82	12	0.76	12
	*NbEF1α*	0.62	11	0.61	10	4.61	0.78	13	0.72	10
	*NbPGK*	0.50	8	0.22	2	2.65	0.53	10	0.56	2
	*NbTspan*	0.13	1	0.48	6	0.64	0.14	1	0.64	4
	*NbRdR6*	0.54	9	0.54	8	2.41	0.58	8	0.72	9
	*NbF-BOX*	0.13	1	0.42	3	0.85	0.20	2	0.61	3
	*NbUBC*	0.19	3	0.59	9	0.98	0.21	3	0.70	8
	*NbACT*	0.47	7	0.10	1	2.09	0.38	5	0.52	1
	*NbAGO2*	0.23	4	0.66	11	1.17	0.30	4	0.76	11
	*NbGAPDH*	0.58	10	0.47	5	3.21	0.61	11	0.66	6
PoPeVYV	*NbeIF4A*	0.10	1	0.15	3	1.22	0.25	7	0.26	4
	*NbLip*	0.24	10	0.31	9	1.17	0.34	6	0.37	9
	*NbL23*	0.22	9	0.34	12	1.91	0.40	13	0.38	10
	*NbKLC*	0.27	11	0.32	10	1.08	0.25	5	0.39	12
	*NbEF1α*	0.15	5	0.17	6	1.53	0.27	12	0.28	6
	*NbPGK*	0.10	1	0.15	5	1.23	0.26	8	0.26	3
	*NbTspan*	0.11	3	0.06	1	0.88	0.19	4	0.24	1
	*NbRdR6*	0.31	13	0.38	13	1.48	0.37	11	0.42	13
	*NbF-BOX*	0.17	6	0.07	2	0.60	0.15	2	0.24	2
	*NbUBC*	0.19	8	0.17	6	0.58	0.13	1	0.28	7
	*NbACT*	0.19	7	0.15	4	0.79	0.15	3	0.27	5
	*NbAGO2*	0.29	12	0.33	11	1.30	0.36	10	0.38	11
	*NbGAPDH*	0.14	4	0.20	8	1.29	0.25	9	0.29	8

M: expression stability value; SV: stability value; CV: coefficient of variation; SD: standard deviation; R: Ranking.

**Table 3 plants-12-00857-t003:** Gene expression stability ranked using RefFinder.

Genes	Tobamovirus					Potyvirus				Potexvirus	Polerovirus
	TMV	PMMoV	TMGMV	ToBRFV	ToMMV	TuMV	ChiRSV	ChiVMV	PePMoV	PVX	PoPeVYV
*NbeIF4A*	2	7	10	7	4	3	6	2	2	7	3
*NbLip*	1	9	3	5	3	7	8	6	1	13	10
*NbL23*	4	11	8	11	10	8	12	11	7	6	11
*NbKLC*	5	6	7	6	7	6	9	8	3	12	9
*NbEF1α*	3	8	6	3	5	5	3	7	7	11	8
*NbPGK*	8	12	5	4	8	1	5	5	6	4	4
*NbTspan*	6	3	9	10	2	4	1	1	4	2	1
*NbRdR6*	7	1	2	8	1	2	4	10	10	10	13
*NbF-BOX*	9	5	1	1	9	9	7	4	5	1	2
*NbUBC*	10	2	11	2	6	10	2	3	9	5	5
*NbACT*	11	4	4	9	12	11	11	9	11	3	6
*NbAGO2*	12	13	12	13	11	12	13	13	13	8	12
*NbGAPDH*	13	10	13	12	13	13	10	12	12	9	7

## Data Availability

The data presented in this article are available on request from the corresponding authors.

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
