# Peer review of "Selection and Validation of Reference Genes for RT-qPCR Analysis of Gene Expression in Nicotiana benthamiana upon Single Infections by 11 Positive-Sense Single-Stranded RNA Viruses from Four Genera"

_plants, 2023, doi:10.3390/plants12040857_

Round 1

Reviewer 1 Report

The manuscript by Zhang and colleagues aims to select, analyze and validate RT-qPCR plant reference genes during viral infection. The manuscript is generally interesting and contains novel information about the selected and analyzed genes upon infection of different viruses from four groups. The idea and the concept of the manuscript promises to reveal very useful practical information. However, in my opinion, authors haven’t reached their goal: they conclude that reference genes should be selected individually for each viral infection and no general and universal reference was suggested. And taking into account that rather often researches deal with simultaneous infection of two viruses/viral vectors, it is important to have arsenal of several most stable genes to check if they are applicable as reference for these conditions.

So my main concern is about the scientific and practical value of this research.

However, I also have some questions on the results and technical details:

-        authors write in the Introduction (page 2) “However, the stability of the expression
of these genes in the conditions of biotic stresses, such as the infection of different viruses,
has not been well studied.” which is actually not true because even in cited papers (for example Liu, 2012) the stability and applicability of the studies genes was thoroughly evaluated upon viral infection. As for assessment of potential reference genes, the paper by Schmidt and Delaney 2010 (DOI: 10.1007/s00438-010-0511-1) should also be included in introduction and discussion

-        authors mention that “TBSV can bind GAPDH,” as a reason why GAPDH mRNA shouldn’t be used as well as other examples when virus interacts with the cellular protein, however, it is not correct to make this conclusion because interaction with the protein doesn’t not always lead to the change in the corresponding mRNA content. So authors should rephrase these statements and check if there is a feedback regulation between particular protein influenced by the virus and its mRNA

-        “Results”, subsection 2.1. “genes in blue letters indicate that Sanger sequencing results were wrong” – what does this phase mean? This is really confusing: the obtained seq didn’t correspond to the expected? or there were mistakes in it? Did you sequence the cloned into plasmids products of RT-PCR? or PCR products per se?

-        in section 2.2. Confirmation of viral infection authors write that the infection was confirmed by symptoms. But I insist on presenting the additional data confirming infection of each of the virus: Coomassie-stained SDS-PAGE where, for example, coat protein band is clearly distinguishable or Western-blot analysis or RT-qPCR results confirming viral reproduction in upper leaves of infected plants. The photos at Fig S2 show symptoms (which are not actually obvious for each group of plants – the photos should be made in different angle and better quality, maybe showing individual leaves with distinct symptoms) only for 4 viruses from the studied 11.

-        the end of 2.3 subsection – correct the repetition of gene names “the
genes with the least variations in Ct values were RdR6, UBC, Lip, and Lip, respectively.”

-        subsection 4.1 – “All viruses were agro-inoculated into N. benthamiana” – do the authors have viral vectors (plasmids), encoding all the viruses? this phrase should be deciphered and explained better. If there are viral vectors then authors should describe them, if authors used viral inoculum then why it is agro-inoculation?

-        4.2. –first lane – should be above, not below in (see Results section 3.2, above)

-        correct the name of Fig S1 and S3 in the list of the figures in the text and below the figures in the Suppl material: Figure S1: Analyssi of RT-qPCR product of 28 genes throug 1%.......... and Figure S3: Analyssi

Reviewer 2 Report

This study is performed in a very steady manner, and the paper is written concisely and easy to read.

I don't have any major issues to this manuscript, and pointed out only two minor issues below.

1: The authors assessed primer specificity using non-infected healthy plant total RNA. Is the primer specificity high even though total RNA from virus-infected plants is used? 

2: Did the authors do DNase treatment after total RNA extraction? I understood that RNA integrity was high by PAGE, but this does not guarantee that RNA does not contain DNA.

Round 2

Reviewer 1 Report

Most of the questions were resolved by the authors, necessary experimental material added. However, there are several points left to fix. See below:

 Paragraph about host-virus interaction (lines 56-76) still suffers from the lack of logic: authors discuss examples when virus interacts with different cellular PROTEIN factors and based on these examples makes a conclusion “Therefore, it is particularly important to select and validate RT-qPCR reference genes to analyze the expression of host genes upon viral infection”. I understand, what they wanted to indicate, but in such a way it is not correct. Authors should refer to examples of effects of viral infection on host gene expression or host RNA stability to make such a statement about accurate selection and validation of reference genes.

Lines 49-53. Paper by Liu et al 2012 must be cited there because most of the commonly used reference genes were evaluated in it at the background of viral infection (Necrovirus (TNV-A and BBSV), Benyvirus (BNYVV), Hordeivirus (BSMV) and Potexvirus) that was not tested in the current manuscript except of Potexvirus. Also the paper by Rotenberg et al 2006 (doi: 10.1016/j.jviromet.2006.07.017) should be mentioned as the suitability of three genes, ubi3, EF-1 and actin was evaluated as internal controls for quantifying gene expression at the background of TRV infection (TRV-based vectors are commonly used for virus-induced gene silencing) in that paper.

Line 65 – better to remove words “plant virus”

Lines 109-110 – How many clones were sequenced? what did the sequences aligned to?

What primers were used to obtain cDNA corresponding to viral genomes for the confirmation of the viral infection?

Authors should check formatting of the Table S3 and make each primer sequence fit a single line. The same is about Table 3 – virus and gene names.
